# Direct and mediated effects of treatment context on low back pain outcome: a prospective cohort study

Felicity Bishop ![ORCID] ,[1] Miznah Al-Abbadey,[1,2] Lisa Roberts ![ORCID] ,[3,4] Hugh MacPherson,[5] Beth Stuart,[6] Dawn Carnes,[7] Carol Fawkes,[7] Lucy Yardley,[1,8] Katherine Bradbury[1]

Deceased

For numbered affiliations see end of article.

**Correspondence to**
Dr Felicity Bishop;
F.L.Bishop@southampton.ac.uk

## ABSTRACT

**Objectives** Contextual components of treatment previously associated with patient outcomes include the environment, therapeutic relationship and expectancies. Questions remain about which components are most important, how they influence outcomes and comparative effects across treatment approaches. We aimed to identify significant and strong contextual predictors of patient outcomes, test for psychological mediators and compare effects across three treatment approaches.

**Design** Prospective cohort study with patient-reported and practitioner-reported questionnaire data (online or paper) collected at first consultation, 2 weeks and 3 months.

**Setting** Physiotherapy, osteopathy and acupuncture clinics throughout the UK.

**Participants** 166 practitioners (65 physiotherapists, 46 osteopaths, 55 acupuncturists) were recruited via their professional organisations. Practitioners recruited 960 adult patients seeking treatment for low back pain (LBP).

**Primary and secondary outcomes** The primary outcome was back-related disability. Secondary outcomes were pain and well-being. Contextual components measured were: therapeutic alliance; patient satisfaction with appointment systems, access, facilities; patients' treatment beliefs including outcome expectancies; practitioners' attitudes to LBP and practitioners' patient-specific outcome expectancies. The hypothesised mediators measured were: patient self-efficacy for pain management; patient perceptions of LBP and psychosocial distress.

**Results** After controlling for baseline and potential confounders, statistically significant predictors of reduced back-related disability were: all three dimensions of stronger therapeutic alliance (goal, task and bond); higher patient satisfaction with appointment systems; reduced patient-perceived treatment credibility and increased practitioner-rated outcome expectancies. Therapeutic alliance over task ($\eta_p^2$=0.10, 95% CI 0.07 to 0.14) and practitioner-rated outcome expectancies ($\eta_p^2$=0.08, 95% CI 0.05 to 0.11) demonstrated the largest effect sizes. Patients' self-efficacy, LBP perceptions and psychosocial distress partially mediated these relationships. There were no interactions with treatment approach.

**Conclusions** Enhancing contextual components in musculoskeletal healthcare could improve patient outcomes. Interventions should focus on helping practitioners and patients forge effective therapeutic

### Strengths and limitations of this study

- ► The prospective but non-experimental design requires cautious interpretation regarding possible causal relationships between variables.
- ► 97% of the sample size was achieved (960 vs 986), with higher retention than anticipated (77% vs 70%) and the final sample at follow-up exceeded that required a priori (742 vs 690).
- ► The geographical spread across the UK and the large cohort size suggests that the results may be generalisable beyond this sample.
- ► All variables were measured using self-report questionnaires, potentially introducing social desirability and/or shared-method bias; the former was partially mitigated by patients returning questionnaires directly to the researchers, not their clinicians; the latter was partially mitigated by including patient and practitioner completed questionnaires and separating in time the measurement of predictors and outcomes.

alliances with strong affective bonds and agreement on treatment goals and how to achieve them.

## INTRODUCTION

All healthcare interventions are delivered within a context that may significantly enhance or hinder an intervention's beneficial effects. Furthermore, contextual components, such as features of the environment or the patient's beliefs or 'mind-set',[1 2] may be shared across similar therapies. Therefore, identifying and strengthening beneficial contextual components could offer an efficient means of enhancing the effectiveness of multiple interventions. Low back pain (LBP) was chosen as the condition of interest as it is highly prevalent with considerable burden to individuals, communities, health services and the economy[3] and there is scope and need to improve patient outcomes across multiple non-pharmacological treatments. However, before embarking on intervention development and testing, it is vital to determine which

contextual components should be targeted to produce the greatest likelihood of patient benefit.

Contextual components comprise the broad constellation of psychological, social and environmental factors that act alongside and can interact with core components of interventions. Five domains have been proposed: patient–practitioner relationship; patients' beliefs; practitioners' beliefs; healthcare environment and, for pharmacological interventions, incidental characteristics of treatment such as the colour of medications.[4] Particular components have been associated with improved patient satisfaction and/or clinical outcomes across multiple settings or specifically within non-pharmacological musculoskeletal care. Such components include: positive and empathetic patient–practitioner relationships and a strong patient–practitioner alliance[4–10] (patient–practitioner relationship); patients' outcome expectancies and perceived treatment credibility[9 11–16] (patient beliefs); musculoskeletal practitioners' beliefs about pain[17–22] and outcome expectancies[23] (practitioner beliefs); good organisational environments (eg, collegiality)[24], the physical–sensory environment (eg, music)[25] and the health-care sector[26–28] (environment), although evidence of effects of healthcare sector on patient outcomes is mainly qualitative. Contextual components might impact patient outcomes via (1) behavioural pathways (eg, improved self-management),[29 30] but see also Haanstra *et al* who found that adherence accounted for only a small proportion of the relationship between expectancies and pain outcomes,[31] (2) psychosocial pathways (eg, reconceptualising pain as less threatening[32–34]; increased self-efficacy[35 36]) and/or (3) neurophysiological pathways.[37–41] The focus in this study was on psychosocial pathways; more details on how these were hypothesised to operate are provided in our published protocol.[42]

In summary, there is good evidence that contextual components can impact patient outcomes. However, most studies have focused on one or two components, meaning that direct comparisons have not been made to ascertain the relative contribution of different contextual components to patient health outcomes. Few studies have quantitatively tested potential pathways in the context of usual clinical practice. Most studies have focused on one treatment, meaning that direct comparisons have not been made to compare contextual components and their effects across treatment approaches. Therefore, the purpose of this study was to investigate the effects and mediating pathways of multiple contextual components on patient outcomes across three different treatment approaches in the private and public sectors. Physiotherapy, osteopathy and acupuncture were selected as they are popular, safe and relatively effective approaches to treating LBP[43–45] and the core treatment modalities used by physiotherapists, osteopaths and acupuncturists were recommended by clinical guidelines at the conception of this study.[46 47]

The aims of the study were to (1) identify contextual components of treatment for LBP that predict patient outcomes, (2) test whether psychosocial factors mediate effects of contextual components on patient outcomes and (3) compare components and their effects across physiotherapy, osteopathy and acupuncture. In relation to aim (1), based on the literature it was hypothesised that patients would experience less back-related disability after treatment for LBP when contextual components are more positive (ie, stronger therapeutic alliance; more pleasant/accessible/convenient/supportive healthcare environment; positive patient-reported treatment beliefs and outcome expectancies; positive practitioner-reported outcome expectancies and practitioner having a biopsychosocial orientation to LBP). However, there were no strong grounds for a priori hypotheses concerning which component would be the strongest predictor of patient outcomes. In relation to aim (2), it was hypothesised that contextual components reduce patients' back-related disability by reducing the perceived threat of pain and/or increasing patients' self-efficacy for coping with pain and/or reducing psychosocial distress. For aim (3), the effects of contextual components on patient outcomes were hypothesised to be larger in osteopathy and acupuncture—as examples of what have been classed complementary therapies—than in physiotherapy, which is usually considered a conventional therapy. This hypothesis was based on evidence that patients find the context of complementary therapies particularly valuable,[48] that the consultation process used by acupuncturists is effective for irritable bowel syndrome even when accompanied by a sham version of acupuncture[8] and that the rituals and modes of delivery of complementary therapies may enhance their effects.[49]

## METHODS
### Design
In this prospective cohort study, practitioners and patients completed questionnaires at three time-points: baseline (after the first consultation for a new episode of LBP); during treatment (2 weeks post-baseline) and short-term outcome (3 months post-baseline). Contextual components were measured once each either at baseline or 2 weeks (specified in the Measures section), with time-points chosen to reduce ceiling effects, capture data accurately and spread questionnaire burden. Outcomes, potential predictors of outcome and potential mediators were measured repeatedly at baseline, 2 weeks and 3 months, to permit tests of whether scores on contextual factors were associated with changes over time in predictors or mediators. Participants chose to complete hard copies (mailed, returned via Freepost) or electronic copies (emailed, completed online at isurvey.soton.ac.uk). Online and paper versions of the primary outcome are equivalent, and used interchangeably, as patients value having this choice.[50] The full protocol was published.[42] See online supplemental table 1 for deviations from the protocol.

## Procedure

Between April 2015 and April 2017, physiotherapists, osteopaths and acupuncturists working in the National Health Service (NHS) and the private sector throughout the UK were recruited by advertisements and personal invitations, with support from professional societies. Inclusion criteria were: treat at least one patient with LBP on average per week, have at least 3 years' current experience in musculoskeletal work and be registered (or eligible to register, for acupuncturists) with the appropriate professional body. To better capture current clinical practice, no restrictions were placed on type of intervention being delivered, meaning that osteopaths and physiotherapists were able to use acupuncture if and when they would usually do so.

Written consent was obtained from eligible practitioners, who gave potentially eligible patients a study invitation pack (containing invitation letter, information sheet, consent form and baseline questionnaire). Patients were recruited between May 2015 and April 2017. On inviting an eligible patient into the study, practitioners rated their outcome expectancy for that patient on an anonymised form, labelled only with the same unique identifier as the corresponding patient invitation pack. Each outcome expectancy form was only analysed if the corresponding patient consent to participate was received by the researchers. Inclusion criteria were: at least 18 years old, seeking treatment from a participating practitioner at their first consultation for a new episode of LBP and score at least 4/24 on the Roland-Morris Disability Questionnaire (RMDQ).[51] Exclusion criteria were: unable to complete questionnaires in English or Welsh, have serious underlying pathology (inflammatory arthritis, malignancy) or have practitioner-identified conditions that would prevent the sought treatment being applied. Arguably, LBP as managed in primary care should be conceptualised as typically having a chronic-episodic timeline with recurrent acute flare-ups against a backdrop of temporary remission or less bothersome symptoms.[52 53] Broad inclusion criteria therefore best reflect the clinical situation.

Completed consent forms and questionnaires were returned directly to the researchers via prepaid envelopes. The researchers emailed or posted (participant's choice) the 2 weeks and 3 months questionnaires to participants. Techniques used to enhance retention were: non-response follow-ups (resending questionnaires and telephoning); monthly study e-newsletters to practitioners; small gifts (eg, tea bag/pen) and monetary incentives (£5 voucher) for patients; personalised questionnaires; ink-signed cover letters; coloured ink; stamped return envelopes; first class post.[54]

## Measures

This section describes how each construct was operationalised, explains the choice of questionnaires and the timing of each measure. Where other measures of constructs existed, questionnaires were chosen for their conceptual fit, psychometric properties and conciseness. Online supplemental table 2 provides additional details including: example items, response scales and Cronbach's alphas.

## Outcomes

The primary outcome was self-reported back-related disability, measured using the 24-item RMDQ.[51] Secondary outcomes (pain intensity, well-being, work and social role disability, satisfaction with care) were measured using recommended core single items.[55] All primary and secondary outcomes were measured at baseline, 2 weeks and 3 months.

## Contextual components

### Patient–practitioner relationship

The patient–practitioner relationship was operationalised in this study as the therapeutic alliance. The therapeutic alliance is conceptually grounded in theory on how patient–practitioner interactions can elicit psychological and/or behavioural changes[56] both of which may be important in LBP. Therapeutic alliance was assessed using the 12-item patient-reported Working Alliance Inventory Short Form (WAI-SF), which measures three dimensions of working alliance—task collaboration, affective bond and goal concordance—(four items each) with acceptable psychometric properties.[57] Task collaboration refers to the extent to which patients feel that the treatment approach their practitioner is taking is the right one for them. Affective bond refers to the extent to which patients feel an interpersonal connection with their practitioner including mutual respect and liking. And goal concordance refers to the extent to which patients feel they agree with their practitioner on the outcomes they are aiming to achieve through treatment. The patient-reported WAI-SF may have ceiling effects if used after one treatment but tends to remain stable after the second treatment[6]; it was therefore administered at 2 weeks.

### The healthcare environment

Practitioners' perceptions of the organisational environment were measured at baseline using two subscales from the psychometrically sound Attitudes to Back Pain Scale—Musculoskeletal Practitioners (ABS-mp): putting limits on sessions (four items) and perceived connections within the healthcare system (three items).[17 58] Patients' perceptions of the organisational and sensory–physical environment were assessed at 2 weeks using three subscales of the Patient Satisfaction Questionnaire designed to assess patient perceptions of the quality of primary healthcare in the UK.[59] The access subscale measures perceptions of interactions with reception staff (eight items); the appointments subscale measures perceived availability of convenient appointments (four items); the facilities subscale measures perceptions of the physical environment of the clinic and waiting room (four items).

### Patients' beliefs

Four dimensions of treatment beliefs were measured using the brief LBP Treatment Beliefs Questionnaire: expected effectiveness, credibility, concerns and individualised fit (four items per subscale).[60] This questionnaire was explicitly designed for use in mixed cohorts of patients with LBP undergoing diverse treatments. It was completed at baseline as patients' expectancies regarding effectiveness should be measured early in treatment.[13]

### Practitioners' beliefs

A single-item numerical rating scale at baseline measured practitioners' outcome expectancies for each patient. Four subscales from the ABS-mp measured, at baseline, practitioners': willingness to engage with psychological issues (four items); confidence and concern over clinical limitations (two items); re-activation of work and activity (three items) and belief in an underlying structural cause of pain (three items).[17 58]

### Hypothesised mediators

Hypothesised mediators, that is, constructs thought to be on the causal pathway between contextual components and patient outcomes, were measured at baseline, 2 weeks and 3 months. The reliable and valid 9-item Brief Illness Perceptions Questionnaire was used to measure the extent to which patients perceive their LBP as threatening.[61] Self-efficacy for coping with pain was assessed using the 5-item Chronic Pain Self-Efficacy for Pain Management subscale.[62] Psychosocial distress was assessed using the 5-item Psychosocial scale on the STarT Back Questionnaire[63] which assesses multiple modifiable psychosocial predictors of outcome.

### Clinical and sociodemographic characteristics

Patient and practitioner characteristics were assessed at baseline. Patient characteristics were: leg pain and shoulder/neck pain bothersomeness (measured using the single item on bothersomeness as worded on the STarT Back tool[63] adapted from Dunn and Croft[64]); LBP duration; age; gender; work status; compensation status; comorbidities; co-treatments; socioeconomic status. To indicate socioeconomic status, patients' postcodes were collected and used to assign the associated index of multiple deprivation (IMD) score to assign to each patient. The IMD ranks 32 844 small areas in England from most to least deprived; scores from 1 to 10 are then assigned to each area indicating the decile of deprivation, and these can be looked up by postcode. An IMD score of 1 indicates that the area is among the most deprived 10% of areas.[65] Practitioner characteristics were: time since qualifying; experience in musculoskeletal care; age; gender.

Practitioners reported for each patient individually at 3 months which treatment modalities had been applied (reported in online supplemental table 3).

### Statistical analysis

#### Power calculation

The planned multilevel analysis required a final sample size of 690 patients to detect correlations between the predictors, hypothesised mediators and the primary outcome with Pearson's R=0.2 (and corresponding Beta regression coefficients of 0.235) with 95% power and p<0.0005 (two-tailed). This sample size was adjusted for clustering of patients within practitioners: the basic sample size was inflated by a factor of 1+(K–1)*ICC, where K=8 and ICC=0.010[66] (K=mean patients recruited per practitioner, ICC=intra-cluster correlation). It was also intended to allow testing for interactions and possible treatment-specific effects. The target sample size required 86 practitioners to actively recruit patients (29 practitioners per therapy). To allow for 30% dropout, based on a similar prospective cohort study in acupuncture,[15] we aimed to recruit 986 patients.

#### Data entry

Paper questionnaires were entered online by researchers and checked for accuracy. All data were downloaded into MS Excel, cleaned and exported into SPSS V.22 and Stata for analysis.

#### Analysis

Data were explored using descriptive statistics and graphs. Cronbach's alpha was calculated to summarise the internal consistency of the measures at baseline. No imputation was undertaken for missing values.

To test the extent to which each contextual component predicted patient outcomes over time (aim 1), multilevel linear mixed models were run with observations nested within patients over time and patients nested within practitioners. Separate models were run to test the effect of each individual contextual component on patient outcomes. These models excluded hypothesised mediators but did control for baseline back-related disability and other potential confounders specified in the protocol: leg pain and shoulder/neck pain bothersomeness; duration of LBP; age; gender; work status; compensation status; comorbidities; co-treatments; socioeconomic status (using deprivation index as indicated by postcode). The effect size was calculated for significant predictors using the partial eta-squared for multivariate regression, which can be interpreted as follows: 0.02 is a small effect, 0.13 moderate and 0.26 large.[67]

To test whether the effect of contextual components on patient outcomes was mediated by perceived threat of pain, self-efficacy for coping and/or psychosocial distress (aim 2), the Baron and Kenny approach was used.[68] This involved (1) testing whether each hypothesised mediator was associated with the outcome (RMDQ), and then (2) controlling for those hypothesised mediators that were associated with the outcome in the analyses of those contextual components which had a statistically significant relationship with the outcome. This enabled a determination of whether the relationship between the

contextual components and outcome remained significant or whether the relationship was largely explained by the presence of the mediators. Mediators were included in models as repeated measures.

To compare the effects of contextual components on patient outcomes across osteopathy, acupuncture and physiotherapy (aim 3), the presence of significant interactions between contextual components and treatment type was tested.

## Patient public involvement

One volunteer, with personal experience of back pain and clinical research, acted as a patient advisor in this study. She attended team meetings and was involved in decision-making around study procedures and conduct including advising on the design of patient-facing study documents. She also wrote to participants to thank them for taking part. Involving members of the public in the management of research helps to improve researchers' awareness of participants' needs and perspectives and can enhance the design and conduct of studies.[69]

## RESULTS

### Participants

Overall, 166 practitioners and 960 patients participated. On average, each practitioner recruited 5.7 patients (SD=8.2). Figure 1 shows overall participant flow; this is split by treatment group in online supplemental figure 1.

Table 1 shows patient characteristics at baseline, split by treatment group. Most patients were women, university-educated, resided in the 50% least deprived neighbourhoods in the country, had LBP for up to 3 months, did not have a significant comorbidity and were using additional treatment(s). The group of patients accessing physiotherapy on the NHS differed from the other groups, they were less likely to hold postgraduate qualifications, more likely to be unemployed or on sick leave, more likely to have chronic LBP and more likely to report a compensation or other legal claim related to their LBP (absolute numbers of claims were low).

As a cohort, practitioners had been working for 20 years since qualifying (M=20.6, SD=10.1) and felt they were highly experienced at treating patients with LBP (M=4.3, SD=0.8). Compared with the other groups, physiotherapists working in the NHS had been working for fewer years since qualifying (M=15.5, SD=7.2) and rated themselves as less experienced with LBP (M=3.8, SD=1.3). Online supplemental table 4 presents full practitioner characteristics by treatment group.

### Outcomes over time

On average, patients improved from baseline to 2 weeks and from 2 weeks to 3 months on all primary and secondary outcome measures, except for overall satisfaction with healthcare which remained similarly high at all measurement points. Figure 2 displays the change in back-related disability over time; outcomes by treatment

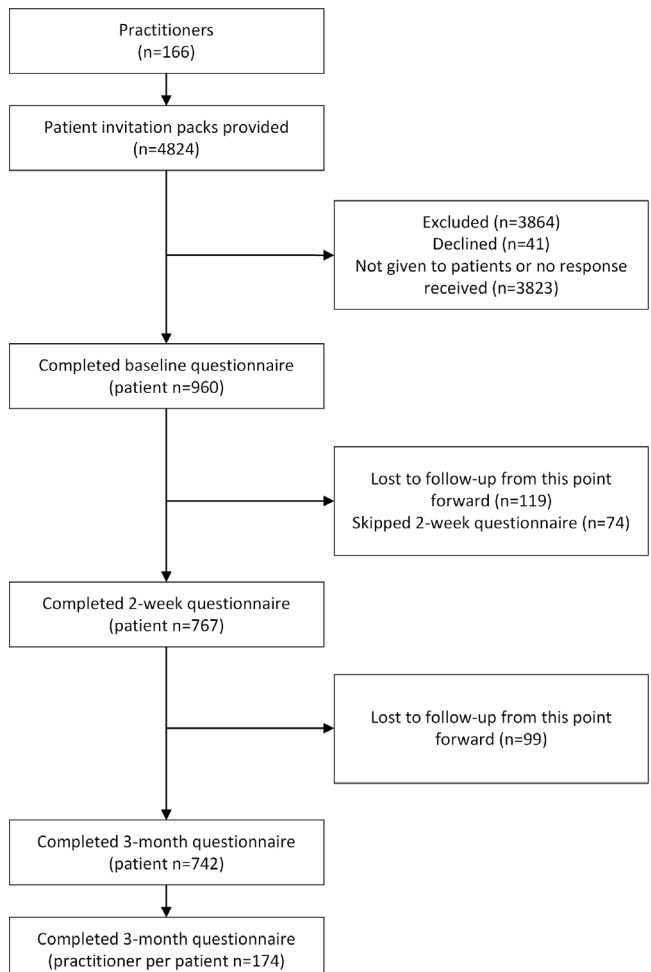

**Figure 1** Modified consort diagram. Showing participant flow, recruitment and drop-out.

group and secondary outcomes are provided in online supplemental figure 2 and table 5, respectively.

### Contextual predictors of back-related disability over time

Table 2 shows the results of models assessing the relationship between each contextual component and back-related disability over time. After controlling for baseline disability and potential confounders, statistically significant predictors of reduced back-related disability over time were: stronger therapeutic alliance on all three dimensions; higher patient satisfaction with appointment systems; reduced perceived credibility of chosen treatment and increased practitioner-rated outcome expectancies.

Therapeutic alliance concerning task and practitioner-rated outcome expectancies demonstrated the largest effect sizes. For each unit increase in patients' rating of the therapeutic alliance (task), patient-rated back-related disability over time decreased by 2.33 points (95% CI −2.89 to −1.77) on the RMDQ. For each unit increase in practitioners' patient-specific outcome expectancies, patient-rated back-related disability over time decreased by 1.29 points (95% CI −1.66 to −0.92) on the RMDQ.

**Table 1** Characteristics at baseline of patients with a new episode of low back pain, for the whole sample and for subgroups of patients receiving physiotherapy in the NHS, physiotherapy in the private sector, osteopathy and acupuncture, n (%)

| | Whole sample | Physiotherapy NHS | Physiotherapy private sector | Osteopathy | Acupuncture |
|---|---|---|---|---|---|
| *Practitioners* | | | | | |
| Total | 166 (100) | 36 (21.7) | 29 (17.5) | 46 (27.7) | 55 (33.1) |
| *Patients* | | | | | |
| Total | 960 (100) | 196 (20.4) | 165 (17.2) | 394 (41.0) | 205 (21.4) |
| Age (years): M (SD) | 52.3 (14.7) | 52.4 (15.2) | 55.4 (14.6) | 57.3 (14.3) | 56.0 (15.0) |
| Gender | | | | | |
| Female | 605 (63.0) | 127 (64.8) | 103 (62.4) | 238 (60.4) | 137 (66.8) |
| Male | 338 (35.2) | 65 (33.2) | 61 (37.0) | 151 (38.3) | 61 (29.8) |
| Missing data | 17 (1.8) | 4 (2.0) | 1 (0.6) | 5 (1.3) | 7 (3.4) |
| Education | | | | | |
| Up to secondary | 39 (4.1) | 12 (6.1) | 1 (0.6) | 18 (4.6) | 8 (3.9) |
| Finished secondary | 306 (31.9) | 72 (36.73) | 39 (23.6) | 128 (32.5) | 67 (32.7) |
| Finished sixth form | 214 (22.3) | 51 (26.0) | 35 (21.2) | 94 (23.9) | 34 (16.6) |
| Undergraduate | 258 (26.9) | 40 (20.4) | 62 (37.6) | 97 (24.6) | 59 (28.8) |
| Postgraduate | 123 (12.8) | 13 (6.6) | 27 (16.4) | 51 (12.9) | 32 (15.6) |
| Missing data | 20 (2.1) | 8 (4.1) | 1 (0.6) | 6 (1.5) | 5 (2.4) |
| Work status | | | | | |
| Employed | 527 (54.9) | 116 (59.2) | 92 (55.8) | 211 (53.6) | 108 (52.7) |
| Unemployed/sick leave | 65 (6.8) | 22 (11.2) | 8 (4.8) | 22 (5.6) | 13 (6.3) |
| Student/homemaker | 60 (6.3) | 11 (5.6) | 14 (8.5) | 20 (5.1) | 15 (7.3) |
| Retired | 280 (29.2) | 43 (21.9) | 50 (30.3) | 125 (31.7) | 62 (30.2) |
| Missing data | 28 (2.9) | 4 (2.0) | 1 (0.6) | 16 (4.1) | 7 (3.4) |
| Healthcare sector | | | | | |
| Private sector | 742 (77.3) | 0 | 165 (100) | 389 (98.7) | 188 (91.7) |
| NHS | 218 (22.7) | 196 (100) | 0 | 5 (1.3) | 17 (8.3) |
| Duration of back pain | | | | | |
| Up to 1 week | 127 (13.2) | 8 (4.1) | 37 (22.4) | 67 (17.0) | 15 (7.3) |
| 1 week to 1 month | 257 (26.8) | 20 (10.2) | 50 (30.3) | 140 (35.5) | 47 (22.9) |
| 1–3 months | 179 (18.6) | 41 (20.9) | 33 (20.0) | 77 (19.5) | 28 (13.7) |
| 3 months to 1 year | 158 (16.5) | 57 (29.1) | 23 (13.9) | 48 (12.2) | 30 (14.6) |
| 1 year+ | 214 (22.3) | 62 (31.6) | 20 (12.1) | 55 (14.0) | 77 (37.6) |
| Missing data | 25 (2.6) | 8 (4.1) | 2 (1.2) | 7 (1.8) | 8 (3.9) |
| Number of previous episodes | | | | | |
| None | 210 (21.9) | 58 (29.6) | 33 (20.0) | 64 (16.2) | 55 (26.8) |
| 5 or fewer | 317 (33.0) | 62 (31.6) | 60 (36.4) | 147 (37.3) | 48 (23.4) |
| More than 5 | 349 (36.4) | 53 (27.0) | 62 (37.6) | 154 (39.1) | 80 (39.0) |
| Missing data | 84 (8.8) | 23 (11.7) | 10 (6.1) | 29 (7.4) | 22 (10.7) |
| Comorbidity | | | | | |
| Longstanding illness or disability (besides LBP) | 321 (33.4) | 71 (36.2) | 51 (30.9) | 119 (30.2) | 80 (39.0) |
| Missing data | 17 (1.8) | 7 (3.6) | 1 (0.6) | 8 (2.0) | 5 (2.4) |
| Co-treatment | | | | | |
| Currently using other treatments | 562 (58.5) | 123 (62.8) | 89 (53.9) | 215 (54.6) | 135 (65.9) |
| Missing data | 17 (1.8) | 10 (5.1) | 2 (1.2) | 12 (3.0) | 4 (2.0) |

Continued

**Table 1** Continued

| | Whole sample | Physiotherapy NHS | Physiotherapy private sector | Osteopathy | Acupuncture |
|---|---|---|---|---|---|
| Compensation | | | | | |
| Compensation or legal claim for LBP | 27 (2.8) | 13 (6.6) | 5 (3.0) | 6 (1.5) | 3 (1.5) |
| Missing data | 17 (1.8) | 4 (2.0) | 1 (0.6) | 6 (1.5) | 4 (2.0) |

LBP, low back pain; NHS, National Health Service.

## Mediation analysis to examine mechanisms

All the mediators were significantly associated with the outcome (for descriptive statistics see online supplemental table 6). For each unit increase in self-efficacy for coping with pain, RMDQ improved by 0.07 (95% CI −0.08 to −0.05). As psychosocial distress increased by one unit, RMDQ increased by 1.43 (95% CI 1.09 to 1.77). And as overall perception of back pain as threatening increased by one unit, RMDQ Score increased by 0.23 (95% CI 0.21 to 0.25). After controlling for these three variables in the longitudinal multilevel models, the measures of therapeutic alliance, satisfaction with appointment systems and reduced perceived credibility of treatment were no longer statistically significant predictors of RMDQ (table 2). This suggests that their effects on disability are indeed mediated by increased self-efficacy for coping with pain, reduced perception of back pain as threatening and reduction in psychosocial distress.

Practitioners' outcome expectancies at baseline showed reduced but still statistically significant effects on RMDQ changes over time, suggesting that their effects are only partially mediated. In all cases the mediators remained associated with the outcome in the multivariable models, except for self-efficacy, which was not statistically significant in the model with practitioners' outcome expectancies. This suggests that the effect of practitioners' outcome expectancies at baseline on changes in RMDQ

over time was partially mediated by changes in perception of back pain as threatening and psychosocial distress. For a visual summary of the general mediation model tested, see online supplemental figure 3.

## Comparisons across treatments

The longitudinal models revealed no significant interactions between contextual factors and treatment type (see online supplemental table 7). This indicates that the effects of contextual components on back-related disability do not differ significantly between osteopathy, acupuncture and physiotherapy.

## DISCUSSION

This large prospective cohort study investigated the effects and mediating pathways of multiple contextual components on patient outcomes across three different therapies in the private and public sectors. As hypothesised, components representing four non-pharmacological contextual domains were significantly associated with back-related disability over time in a large cohort of patients with LBP seeking osteopathy, acupuncture and physiotherapy. The patient–practitioner therapeutic alliance and practitioners' expectancies of how individual patients will respond to treatment were the strongest contextual components, showing the largest effects on patient outcomes. These components reduced disability at least partly by improving patients' self-efficacy for coping and reducing psychosocial distress and the perceived threat of LBP. Other contextual components emerged as statistically significant predictors of improved outcome over time: the healthcare environment, specifically greater satisfaction with appointments; and (contrary to hypothesised direction of association) patients' beliefs, specifically reduced perceived credibility of one's chosen treatment. The relationship between contextual components and patient outcomes did not differ across osteopathy, acupuncture and physiotherapy.

The current study extends understanding of the mechanisms through which contextual factors may impact patient health outcomes. Earlier studies have investigated pathways among smaller numbers of contextual variables and suggested, for example, that self-efficacy mediates the impact on osteoarthritis pain of practitioners communicating positive expectancies,[35] affective aspects of the therapeutic relationship may shape

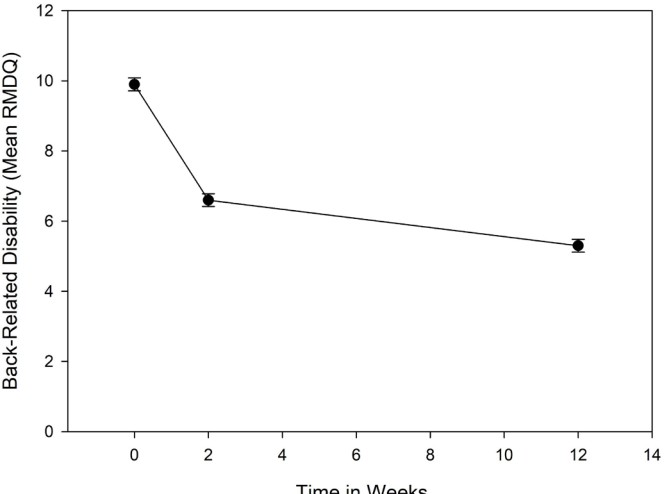

**Figure 2** Back-related disability over time. Mean scores and SEs are plotted. RMDQ, Roland-Morris Disability Questionnaire.

**Table 2** Longitudinal models assessing contextual predictors of back-related disability over time in patients receiving physiotherapy, osteopathy or acupuncture for a new episode of low back pain

| Contextual component | Possible scale range | Mean (SD) | Change in RMDQ for one-unit change in component | | | |
| --- | --- | --- | --- | --- | --- | --- |
| | | | Unadjusted for confounders | Adjusted for confounders* | Adjusted for mediators† | Effect size partial eta-squared (95% CI)‡ |
| Therapeutic alliance: goal | 1–5 | 4.2 (0.7) | −1.14 (−1.69 to −0.60) | −0.94 (−1.54 to −0.35) | 0.36 (−0.09 to 0.82) | 0.02 (0.004 to 0.04) |
| Therapeutic alliance: task | 1–5 | 4.0 (0.6) | −2.78 (−3.33 to −2.23) | −2.33 (−2.89 to −1.77) | 0.13 (−0.34 to 0.61) | 0.10 (0.07 to 0.14) |
| Therapeutic alliance: bond | 1–5 | 4.1 (0.9) | −1.14 (−1.56 to −0.32) | −0.86 (−1.30 to −0.43) | −0.10 (−0.43 to 0.23) | 0.03 (0.01 to 0.05) |
| Environment: limits on treatment | 4–28 | 11.5 (2.9) | −0.08 (−0.25 to 0.09) | −0.03 (−0.23 to 0.16) | | |
| Environment: connections within the healthcare system | 3–21 | 11.3 (2.3) | 0.01 (−0.23 to 0.25) | −0.05 (−0.27 to 0.17) | | |
| Environment: satisfaction with appointment systems | 1–5 | 1.9 (0.7) | 1.31 (0.81 to 1.82) | 0.67 (0.13 to 1.20) | −0.23 (−0.65 to 0.14) | 0.02 (0.01 to 0.04) |
| Environment: satisfaction with access | 1–5 | 2.5 (0.6) | 1.01 (0.38 to 1.63) | 0.60 (−0.04 to 1.26) | | |
| Environment: satisfaction with facilities | 1–5 | 2.2 (0.8) | 0.52 (0.04 to 1.01) | 0.34 (−0.18 to 0.86) | | |
| Patient's beliefs: perceived credibility of treatment | 1–5 | 3.0 (0.3) | 1.34 (0.19 to 2.50) | 1.51 (0.35 to 2.67) | 0.58 (−0.28 to 1.44) | 0.01 (0.001 to 0.03) |
| Patient's beliefs: expectancies for treatment effectiveness | 1–5 | 2.8 (0.3) | 0.17 (−0.99 to 1.34) | 0.12 (−1.05 to 1.29) | | |
| Patient's beliefs: concerns about treatment | 1–5 | 2.9 (0.3) | −0.36 (−1.45 to 0.73) | −0.26 (−1.37 to 0.86) | | |
| Patient's beliefs: individualised fit of treatment | 1–5 | 2.9 (0.3) | 0.14 (−1.07 to 1.34) | 0.59 (−0.65 to 1.83) | | |
| Practitioner's beliefs: psychological | 4–28 | 17.8 (3.6) | 0.13 (−0.06 to 0.32) | 0.03 (−0.15 to 0.20) | | |
| Practitioner's beliefs: confidence | 2–14 | 8.2 (2.0) | 0.03 (−0.27 to 0.32) | 0.10 (−0.19 to 0.40) | | |
| Practitioner's beliefs: reactivation | 3–21 | 15.3 (3.0) | −0.06 (−0.24 to 0.13) | 0.08 (−0.10 to 0.25) | | |
| Practitioner's beliefs: biomedical | 3–21 | 12.2 (3.6) | −0.06 (−0.22 to 0.10) | −0.07 (−0.25 to 0.11) | | |
| Practitioner's beliefs: outcome expectancies | 1–7 | 5.7 (1.1) | −1.77 (−2.12 to −1.42) | −1.29 (−1.66 to −0.92) | −0.42 (−0.70 to −0.13) | 0.08 (0.05 to 0.11) |

Table reports the results of separate models run to test the effect of each individual contextual factor on patient outcomes, specifically back-related disability.
*Adjusted for baseline back-related disability and: leg pain and shoulder/neck pain bothersomeness; duration of LBP; age; gender; work status; compensation status; comorbidities; co-treatments; socioeconomic status.
†Adjusted for mediation by perceived threat of pain, self-efficacy for coping and psychosocial distress.
‡Effect size only reported for statistically significant contextual predictors of back-related disability over time.
RMDQ, Roland Morris Disability Questionnaire.

patient expectancies in chronic pain,[70] and expectancies might impact pain via reduced catastrophising in chronic back pain.[33] The current results further suggest contextual factors reduce back-related disability in part by enhancing patient self-efficacy for coping with pain, reducing the perceived threat of pain and reducing psychosocial distress. This is broadly consistent with the fear avoidance model[34] and social cognitive theory.[36] It also builds on work in adjacent areas such as communication[41] and starts to map potential evidence-based psychological pathways through which contextual factors might impact patient outcomes.

The prospective but non-experimental design requires cautious interpretation regarding possible causal relationships between variables: the results provide evidence of predictive relationships between variables over time, which are consistent with, but insufficient to establish causality. Experimental designs (eg, trials) are now required to build on this work and to confirm the clinical importance of therapeutic alliance as a factor that influences patient outcomes. While 97% of the baseline sample size was achieved (960 vs 986), retention was higher than anticipated (77% vs 70%) and the final sample at follow-up exceeded that required a priori (108%, n=742 vs 690). The geographical spread across the UK and the large cohort size suggests that the results may be generalisable beyond this sample. All variables were measured using self-report questionnaires, potentially introducing social desirability and/or shared-method bias; the former was partially mitigated by patients returning questionnaires directly to the researchers, not their clinicians; the latter was partially mitigated by the inclusion of patient and practitioner completed questionnaires and the separation in time of the measurement of predictors and outcomes. Some of the ABS-MP subscales had low internal consistency in this sample (Cronbach's alphas <0.6) suggesting the results of those analyses may be unreliable. Empathy and perceived competence, potentially important contextual components associated with the therapeutic relationship,[71] were not directly captured by the measure of therapeutic alliance.

The current findings highlight the relative importance of the therapeutic alliance as a contextual predictor of back-related disability over time, compared with other contextual components. The minimum clinically important difference for primary care patients with LBP has been calculated as a standard 2–3 points[72] or a 30% reduction in RMDQ score from baseline.[73] Therefore, a one unit increase in patients' rating of the therapeutic alliance (task) and a two unit increase in practitioners' outcome expectancies would be associated with a clinically important reduction in back-related disability for patients scoring 7 or less on the RMDQ at baseline. For patients scoring 8–15 points on the RMDQ at baseline, a two-unit increase in patients' rating of the therapeutic alliance (task) and a three-unit increase in practitioners' outcome expectancies may be needed to make a clinically-important difference.

Our findings support a growing emphasis on patient–practitioner communication and relationships in musculoskeletal healthcare interactions. Previous studies and reviews have also suggested the importance of the therapeutic relationship for positive patient outcomes[7 10] but have not considered the extensive selection of other contextual variables measured here. A recent systematic review in physiotherapy concluded that there was a lack of evidence of a strong relationship between therapeutic alliance and reduced pain, but the studies reviewed were all much smaller than the current cohort (n=12–182) and may have been underpowered.[74] The only other contextual factor to approach a moderate effect size on patient outcomes was the practitioners' outcome expectancies at baseline, which predicted outcomes even after controlling for baseline clinical and sociodemographic variables. This factor has been somewhat overlooked in the literature but was also found to predict outcomes in a large acupuncture study.[23] The reasons for this association likely involve practitioners making global clinical judgements that incorporate factors not measured in the current study. The statistically significant association between lower perceived treatment credibility and improved patient outcomes was not in the predicted direction; the effect size was very small (0.01) and unlikely to be clinically significant. It may be that patients who believe their treatment is extremely credible are more likely to then experience some disappointment when experiencing the reality of that treatment, which may then impact engagement with treatment and/or outcome reporting. The lack of association between patients' expectancies and subsequent outcomes was unexpected given the balance of previous findings linking patient expectancies to outcomes,[9 75] but not unprecedented: expectancies are more strongly and consistently associated with experimental acute and procedural pain than chronic pain.[11] This null finding may be related to timing, as expectancies may change over time[76] and patients can be reluctant to specify expectancies about effectiveness early in treatment, while concepts such as hope may have greater resonance in clinical settings.[77] The lack of associations between patient outcomes and practitioner perceptions of the healthcare environment, attitudes and orientation to LBP was also unexpected, but may be an artefact of the low reliability of the ABS-mp subscales.

Future research, using qualitative methods, could explore how practitioners form positive outcome expectancies and how this might be harnessed in practice. The possible causal nature of and mechanisms by which practitioner outcome expectancies impact patient outcomes also require further research. To confirm the clinical importance of therapeutic alliance and enable practitioners to harness this in practice, sensible next steps would be to develop and then trial brief and engaging post-qualification training for practitioners, using a systematic approach such as the person-based approach to intervention development.[78]

Having a strong therapeutic alliance and positive practitioner expectancies of outcome were the strongest contextual predictors of reduced disability in a large cohort of patients with LBP receiving physiotherapy, osteopathy and acupuncture. These operate in part by altering known

psychological predictors of outcome: self-efficacy, perceived pain threat and psychosocial distress. Enhancing these contextual components in practice could improve patient health outcomes from LBP. While a strong therapeutic alliance concerning the ultimate goals of treatment and a strong affective bond are associated with improved patient health outcomes, it is the strength of alliance concerning how to achieve the goals (ie, the tasks of treatment) that has the largest effect on improved outcomes. The lack of between-treatment differences in significant predictors suggests the potential for shared learning across therapies to improve contextual components. It is advisable for clinicians to focus on strengthening their alliance building related to how the patient should reach an agreed-upon goal within the context of a strong affective bond. Evidence-based, engaging, training could be developed to support skills in therapeutic alliance building.

**Author affiliations**
[1]Department of Psychology, University of Southampton, Southampton, UK
[2]Department of Psychology, University of Portsmouth, Portsmouth, UK
[3]Health Sciences, University of Southampton, Southampton, UK
[4]Therapy Services, University Hospital Southampton NHS Foundation Trust, Southampton, UK
[5]Health Sciences, University of York, York, UK
[6]Primary Care and Population Sciences, University of Southampton, Southampton, UK
[7]Institute of Population Health Sciences, Queen Mary University of London, London, UK
[8]School of Psychological Science, University of Bristol, Bristol, UK

**Acknowledgements** We would like to acknowledge the vital contribution that three, now deceased, colleagues made to the design and management of this research: Dr Borislav Dimitrov (University of Southampton), Dr Jan Leach (University of Brighton) and Professor George Lewith (University of Southampton). Specifically, George Lewith, Borislav Dimitrov and Jan Leach contributed to conceptualising the study, drafting the protocol and the associated grant application and supervising data collection. We would like to thank all the practitioners and patients who took part in this study. The Clinical Research Network, British Acupuncture Council, Chartered Society of Physiotherapy, General Osteopathic Council and the Institute of Osteopathy assisted with recruitment. Research and administrative assistance was provided by Laura Wilde, Sue Eardley, Maddy Greville-Harris, J Matthew Harvey, Daniel Scanlan, Ricky Wong, Tasnia Priti, Daniela Baramova, Helen Loader, Nicole Collaco, Yasmin Hoque and Senada Merkulaj. Our patient public involvement collaborator was Jo Foster.

**Contributors** FB led the study conception and design and drafted the manuscript. FB, KB, HM, LY and LR conceptualised the study and drafted the protocol and the associated grant application. FB, KB, HM, LY, MA-A, DC, CF and LR drafted the manuscript. MA-A was the research fellow on the study, responsible for data collection under the supervision of FB and KB and in collaboration with all authors. BS undertook the statistical analysis. All authors revised the manuscript critically for important intellectual content and gave final approval of the version to be published.

**Funding** This work was supported by Arthritis Research UK Special Strategic Award grant number 20552.

**Competing interests** FB received an honorarium and travel expenses for presenting the preliminary results of this research at the Acupuncture Research Resource Centre research symposium and has also received speaker's fees and travel expenses from the Acupuncture Association of Chartered Physiotherapists. The authors declare that they have no other competing interests.

**Patient consent for publication** Not required.

**Ethics approval** Ethical approval was obtained from NHS Health Research Authority NRES Committee East Midlands—Derby (REF: 14/EM/1113) on 22 August 2014. Participants gave written informed consent; study documents conveyed the right to decline or withdraw without affecting medical care.

**Provenance and peer review** Not commissioned; externally peer reviewed.

**Data availability statement** Deidentified participant data may be requested from the corresponding author, FB (https://orcid.org/0000-0002-8737-6662). Access may only be permitted for the purposes of checking the reported analyses and an analysis plan must be submitted with any request for access to the data. Due to ethical considerations, reuse for any other purpose is not permitted.

**ORCID iDs**
Felicity Bishop http://orcid.org/0000-0002-8737-6662
Lisa Roberts http://orcid.org/0000-0003-2662-6696

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
