## [Reviewer comments · BMJ Open]

ARTICLE DETAILS

TITLE (PROVISIONAL)	Direct and Mediated Effects of Treatment Context on Low Back Pain Outcome: A Prospective Cohort Study
AUTHORS	Bishop, Felicity; Al-Abbadey, Miznah; Roberts, Lisa; MacPherson, Hugh; Stuart, Beth; Carnes, Dawn; Fawkes, Carol; Yardley, Lucy; Bradbury, Katherine

VERSION 1 – REVIEW

REVIEWER	Joshua Zadro University of Sydney, Australia
REVIEW RETURNED	26-Oct-2020

GENERAL COMMENTS	I thank the authors for the opportunity to review this interesting and well-written paper. The authors aimed to investigate whether contextual factors (e.g. therapeutic alliance, outcome expectancies) influenced treatment outcomes in people with low back pain receiving treatment from a physiotherapist, osteopath or acupuncturist. They also investigated whether psychological factors (e.g. self-efficacy, perceptions of LBP, distress) mediated the association between contextual factors and treatment outcomes, and compared effects across the three health professional types. The authors found that stronger therapeutic alliance, higher patient satisfaction with appointment systems, reduced patient-perceived treatment credibility, and increased practitioner-rated outcome expectancies were associated with reduced LBP disability. Patients' self-efficacy, LBP perceptions, and psychosocial distress partially mediated these relationships. However, health professional type did not influence treatment outcomes. There are several strengths to this study. The authors have done a fantastic job at making this paper easy to follow, especially given the number of predictors, mediators and outcomes investigated. Figure 3 and Table 2 are extremely helpful at orienting the reader to all the analyses. The table about deviations from the protocol helps improve transparency. Patient involvement in study design and conduct and a published protocol are other strengths. Below are some suggestions to further improve the manuscript: Introduction -Page 5, Line 59: RE 'see also Haanstra et al': the authors should explain why this study is important rather than leaving it to the reader to look it up. -Page 6, Line 30: Ref 45 does not seem to match the claim. It isn't a trial or systematic review. I also have an issue with calling physiotherapy and osteopathy therapies - they are health
--

	professions. Guidelines recommend treatments, not professions. Please revise this section to reflect evidence for treatments and not professions. -Page 7, line 14: RE: 'in context of a sham treatment': this seems to conflict with the earlier message about effective therapies. If treatments provided by osteopaths and acupuncturists are effective, why are we referring to them as a sham treatment. Maybe I have misunderstood the meaning of this sentence. Either way, please clarify -Overall, the introduction does a good job at setting up the rationale for this study and describing many of the variables that will be investigated. Methods -Page 7, line 50: be consistent with the terms 'prognostic indicator' and 'predictor'. Choose one and be consistent throughout -Power calculation: the authors should explain why they accounted for such as high rate of drop-outs -Page 13, Line 35: please make the external website a reference Results -the authors should include the scale for each predictor and mediator when discussing the impact of a 'one unit' change (and in Table 2) to put the change into context -Page 18, line 48-57: this paragraph would fit better within the discussion. More importantly, the authors should acknowledge that the MCID should be used for between-group differences (not pre-post changes), so may not be applicable in the context of this study -Page 21, 'Comparison across treatments': Is there a table or figure that presents these results? Discussion -I would like to see a brief discussion about future randomised controlled trials that could build on some of the findings from this study -I would also like to see some discussion about why perceived credibility of treatment had a negative impact on patient disability Supplementary material: In Table 5 and 6, the different time points should appear as separate columns
--	--

REVIEWER	Noa Ben Ami Ariel University, Israel
REVIEW RETURNED	31-Oct-2020

GENERAL COMMENTS	I would like to congratulate the authors on this great work! This prospective cohort study investigated the effects and mediating pathways of multiple contextual components on LBP patient outcomes across three different therapies in the private and public sectors. The patient-practitioner therapeutic alliance and practitioners' expectancies were the strongest contextual components. It might reduce disability by improving patients' self-efficacy for coping and reducing psychosocial distress and the perceived threat of LBP. The current study extends understanding of the mechanisms through which contextual factors may impact patient health outcomes. But, it is a prospective study and not an experimental design and this requires caution.
--

	The manuscript is well written and can help improve patient care by improving caregivers' communication skills and the belief that they can help patients.
--	--

VERSION 1 – AUTHOR RESPONSE

Reviewer: 1

Mr. Joshua Zadro, University of Sydney

Comment	Response
Introduction	
-Page 5, Line 59: RE 'see also Haanstra et al': the authors should explain why this study is important rather than leaving it to the reader to look it up.	Added this clarification: but see also Haanstra et al. who found that adherence accounted for only a small proportion of the relationship between expectancies and pain outcomes ³¹
-Page 6, Line 30: Ref 45 does not seem to match the claim. It isn't a trial or systematic review. I also have an issue with calling physiotherapy and osteopathy therapies - they are health professions. Guidelines recommend treatments, not professions. Please revise this section to reflect evidence for treatments and not professions.	Reference 45 is a large scale survey study that provides evidence for the popularity of these therapies, which is one of the three points made in the preceding phrase. We have revised the description of evidence according to your recommendation as follows: "Physiotherapy, osteopathy, and acupuncture were selected as they are popular, safe, and relatively effective approaches to treating LBP ⁴⁴⁻⁴⁶ and the core treatment modalities used by physiotherapists, osteopaths, and acupuncturists, were recommended by clinical guidelines at the conception of this study. ^{47 48} "
-Page 7, line 14: RE: 'in context of a sham treatment': this seems to conflict with the earlier message about effective therapies. If treatments provided by osteopaths and acupuncturists are effective, why are we referring to them as a sham treatment. Maybe I have misunderstood the meaning of this sentence. Either way, please clarify	This has been clarified as: "that the consultation process used by acupuncturists is effective for Irritable Bowel Syndrome even when accompanied by a sham version of acupuncture ^{8...} "
Methods	
-Page 7, line 50: be consistent with the terms 'prognostic indicator' and 'predictor'. Choose one and be consistent throughout	Removed all mentions of prognostic indicators. Replaced with predictors of outcome.

-Power calculation: the authors should explain why they accounted for such as high rate of drop-outs	This was based on a similar study we published previously in acupuncture. Text amended accordingly: "To allow for 30% dropout, based on a similar prospective cohort study in acupuncture, ¹⁵ we aimed to recruit 986 patients."
-Page 13, Line 35: please make the external website a reference	Added.
Results	
-the authors should include the scale for each predictor and mediator when discussing the impact of a 'one unit' change (and in Table 2) to put the change into context	A column with this information has been added to Table 2. We prefer not to also add this to the text, to avoid repetition and to avoid making the text more difficult to follow.
-Page 18, line 48-57: this paragraph would fit better within the discussion. More importantly, the authors should acknowledge that the MCID should be used for between-group differences (not pre-post changes), so may not be applicable in the context of this study	This paragraph has been moved to the discussion section. In addition to reporting the between-group MCID we also have discussed evidence for the MCID in terms of change from baseline.
-Page 21, 'Comparison across treatments': Is there a table or figure that presents these results?	The results have been inserted as Supplemental Material Table 7.
Discussion	
-I would like to see a brief discussion about future randomised controlled trials that could build on some of the findings from this study	Added to page 25 "To confirm the clinical importance of therapeutic alliance and enable practitioners to harness this in practice, sensible next steps would be to develop and then trial brief and engaging post-qualification training for practitioners, using a systematic approach such as the person-based approach to intervention development. ⁷⁸ "
-I would also like to see some discussion about why perceived credibility of treatment had a negative impact on patient disability	Added to page 23: "The statistically significant association between lower perceived treatment credibility and improved patient outcomes was not in the predicted direction; the effect size was very small (.01) and unlikely to be clinically significant. It may be that patients who believe their treatment is extremely credible are more likely to then experience some disappointment when experiencing the

	reality of that treatment, which may then impact engagement with treatment and/or outcome reporting.”
Supplementary material: In Table 5 and 6, the different time points should appear as separate columns	Amended accordingly

VERSION 2 – REVIEW

REVIEWER	Joshua Zadro The University of Sydney, Australia
REVIEW RETURNED	23-Feb-2021

GENERAL COMMENTS	I thank the authors for addressing my comments. I have nothing more to add. Congratulations on a great paper!
--